# Non-Structural Proteins (Nsp): A Marker for Detection of Human Coronavirus Families

**DOI:** 10.3390/pathogens12091185

**Published:** 2023-09-21

**Authors:** María Concepción Tamayo-Ordóñez, Ninfa María Rosas-García, Benjamín Abraham Ayil-Gutiérrez, Juan Manuel Bello-López, Francisco Alberto Tamayo-Ordóñez, Francisco Anguebes-Franseschi, Siprian Damas-Damas, Yahaira de Jesús Tamayo-Ordóñez

**Affiliations:** 1Laboratorio de Ingeniería Genética, Departamento de Biotecnología, Facultad de Ciencias Químicas, Universidad Autónoma de Coahuila, Saltillo 25280, Coahuila, Mexico; mtamayo@uadec.edu.mx; 2Laboratorio de Biotecnología Ambiental del Centro de Biotecnología Genómica, Instituto Politécnico Nacional, Reynosa 88710, Tamaulipas, Mexico; 3CONAHCYT-Centro de Biotecnología Genómica, Instituto Politécnico Nacional, Biotecnología Vegetal, Reynosa 88710, Tamaulipas, Mexico; 4División de Investigación, Hospital Juárez de México, Ciudad de México 07760, Campeche, Mexico; 5Facultad de Química, Universidad Autónoma del Carmen, Calle 56 N. 4, Av. Concordia Col. Benito Juárez, Ciudad del Carmen 24180, Campeche, Mexicosipriandamas@hotmail.com (S.D.-D.)

**Keywords:** SARS-CoV-2, non-structural proteins, phylogenetic relationship, pangenome analysis, Nsp marker

## Abstract

SARS-CoV-2 was the cause of the global pandemic that caused a total of 14.9 million deaths during the years 2020 and 2021, according to the WHO. The virus presents a mutation rate between 10−5 and 10−3 substitutions per nucleotide site per cell infection (s/n/c). Due to this, studies aimed at knowing the evolution of this virus could help us to foresee (through the future development of new detection strategies and vaccines that prevent the infection of this virus in human hosts) that a pandemic caused by this virus will be generated again. In this research, we performed a functional annotation and identification of changes in Nsp (non-structural proteins) domains in the coronavirus genome. The comparison of the 13 selected coronavirus pangenomes demonstrated a total of 69 protein families and 57 functions associated with the structural domain’s differentials between genomes. A marked evolutionary conservation of non-structural proteins was observed. This allowed us to identify and classify highly pathogenic human coronaviruses into alpha, beta, gamma, and delta groups. The designed Nsp cluster provides insight into the trajectory of SARS-CoV-2, demonstrating that it continues to evolve rapidly. An evolutionary marker allows us to discriminate between phylogenetically divergent groups, viral genotypes, and variants between the alpha and betacoronavirus genera. These types of evolutionary studies provide a window of opportunity to use these Nsp as targets of viral therapies.

## 1. Introduction

The pandemic caused globally by the coronavirus that causes severe acute respiratory syndrome (SARS-CoV-2) emphasizes the ability of human pathogenic coronaviruses (HCoVs) to infect and spread rapidly among the human population due to the lack of understanding of its evolution. Even though, today, technologies have already been created that allow the detection of the virus in the early stages of infection, as well as vaccines that helped us to avoid the replication of the virus within human hosts [1], we must still be aware that the mutation rate of this virus is between 10−5 and 10−3 substitutions per nucleotide site per cell infection (s/n/c) [2];this implies the generation of more virus variants, which, in the future, may become as infectious and deadly as the SARS-CoV-2 variant that caused the global pandemic between the years 2019 and 2022. The general director of the WHO declared in 2023 that COVID-19 continues to represent a global health threat and that it will be important to continue vaccinating vulnerable groups and strengthen surveillance. Recently, the WHO declared that the omicron EG.5 variant known as “eris” has shown increased prevalence, growth advantage, and immune-escape properties. This current scenario regarding COVID-19 exalts the importance of directing current research to search for new molecular markers to detect new strains and even propose new effective therapies and vaccines against the disease.

Due to this, studies aimed at knowing the evolution of this virus could help us to foresee (through the future development of new detection strategies and vaccines that prevent the infection of this virus in human hosts) that a pandemic caused by this virus will be generated again.

The SARS-CoV-2 coronavirus belongs to the family Coronaviridae included in the order Nidovirales of zoonotic origin. This virus is the newest member of a family of coronaviruses that infect humans; four phylogenetically distinct groups (alpha, beta, gamma, delta) have been identified, including seven variants of human coronavirus (HCoVs): alpha-CoVs HCoVs-NL63 and HCoVs-229E, beta-CoVs HCoVs-OC43 and HCoVs-HKU1, the severe acute respiratory syndrome-CoV (SARS-CoV), the Middle East respiratory syndrome-CoV (MERS-CoV), and finally, the severe acute respiratory syndrome coronavirus 2 (SARS-CoV-2), which is a single-stranded RNA virus positive-sense and involved disease-causing COVID-19. The emergence of the latter, identified in Wuhan, China, is responsible for the COVID-19 pandemic; despite the end of the pandemic, efforts continue to be made to understand its molecular evolution of SARS-CoV-2 [1,3].

Coronaviruses (CoV) are a large (~30 kb) single-stranded RNA family which are particularly interesting because they contain the largest genome of naturally occurring stable RNA viruses, encode many gene products, and use a batch synthetic RNA process to produce messenger RNA molecules involved during their viral replication [4,5]. The genome of all coronaviruses contains a basic set of genes: replicase (Rep 1a and 1b), spike (S), envelope (E), membrane (M), and nucleoprotein (N), and a variable number of genes encoding non-structural proteins (Nsp) intercalated between these genes in a position characteristic of each virus group. Genes encoding non-structural proteins (Nsp) are NSP1 to NSP10 and NSP12 to NSP16, encoded by genes located within the 50-region of viral RNA genome [6]. A set of non-structural proteins (Nsp) are generated as cleavage products of the ORF1a and ORF1ab viral polyproteins that assemble to facilitate viral replication and transcription. A key component is the RNA-dependent RNA polymerase (RdRp, known as Nsp12). The production of coronavirus subgenomic messenger RNAs involves the fusion of sequences that are not contiguous in the viral genome. The virion includes the RNA genetic material and structural proteins necessary for invasion of host cells. Once inside the cell, the infecting RNA encodes structural proteins that make up the virus particles; while non-structural proteins (Nsp) direct the assembly, transcription, replication, and control of the virus host and accessory proteins whose function has not been determined [6].

Scientific advances in virus genome sequencing and tools for massive data analysis have greatly contributed to expanding our knowledge about the diversity of CoVs hosted by different vertebrates and have provided CoV sequences from different geographic areas associated with different disease phenotypes. These studies have shown that CoV genomes show great plasticity in terms of genetic content and recombination. Despite these advances, important challenges remain to be addressed; in this sense, non-structural proteins have been studied less.

Derived from the sequence complexity of the genes encoding non-structural proteins (Nsp), it is possible to propose them as therapeutic targets of SARS-CoV-2 or variants for COVID-19, if we know their evolution and the marked differences that can differ between groups of Nsp. In this research, we performed in silico analysis (functional annotation), identification of changes in (non-structural proteins) Nsp domains in the coronavirus genome, and evolutionary relationship of SARS-CoV-2 non-structural proteins (Nsp). The results indicated a marked evolutionary conservation of non-structural proteins that allows us to identify and classify highly pathogenic human coronaviruses into alpha, beta, gamma, and delta groups. The designed Nsp cluster allows us to know the trajectory of SARS-CoV-2 to investigate its recent rapid evolution. Additionally, this evolutionary marker allows us to discriminate between phylogenetically divergent groups, viral genotypes, and variants between the alpha and betacoronavirus genera. These types of evolutionary studies provide a window of opportunity to use these non-structural proteins (Nsp) as targets of viral therapies as scientific progress is being made with RNA-dependent RNA polymerase.

## 2. Materials and Methods

### 2.1. Genomic Analysis and Construction of the Pangenome of Alpha, Beta, Gamma, and Delta Coronaviruses

Alpha, beta, gamma, and delta coronaviruses genomes were used, obtained from the NCBI virus database (https://www.ncbi.nlm.nih.gov/labs/virus/vssi/#/, accessed on 19 February 2023) including seven coronavirus variants (HCoVs-229E, HCoVs-NL63, HCoVs-OC43, HCoVs-HKU1, MERS-CoV, SARS-CoV, SARS-CoV-Tor2, AcCoV-Duck, AcCoV-Turkey). In the alpha and beta coronavirus phylogenetic group, the human pathogenic coronaviruses (HCoVs) were gathered, where SARS-CoV-2, the most recent member of the family of coronaviruses that infect humans, was located.

The construction of the coronavirus pangenome was carried out using the Compute Pangenome v0.0.7 program. The protein families present in the input genomes HCoVs-229E, HCoVs-NL63, HCoVs-OC43, HCoVs-HKU1, MERS-CoV, SARS-CoV, SARS-CoV-Tor2, AcCoV-Duck, AcCoV-Turkey were considered to form the core of the pangenome, while the non-conserved families are accessory proteins.

With the KBase database [7], all the distinct protein families identified in a set of input genomic sequences were listed, as well as the identifiers of the proteins present in each family. This clustering allowed for a powerful set of comparisons to be made, to assess the consistency of functional assignments for highly homologous proteins, and to assess the degree to which each protein family has been conserved across the input set of genomes.

The comparison of each selected genome with the pangenome was performed with the GenomeComparisonSDK v0.0.7 program. This comparison method allowed us to: (I) assess the degree of gene conservation among an input set of genomes, (II) understand how genomes have evolved and adapted to their distinctive environments, and (III) identify novel genes and interesting biology in the context of related genomes. 

### 2.2. Conformation of Tertiary Structures of Nsp Proteins 

CDD was employed for protein sequence annotation based on conserved domain fingerprints and functional sites deduced from these fingerprints. CDD search was performed using a collection of 2738 multiple sequence alignment templates, based on the content of the Pfam and SMART databases. Database search tools were included to support rapid computation of the sequence annotation, as well as domain models imported from a series of databases from external sources (Pfam, SMART, COG, PRK, TIGRFAM) [8].

In order to know the conformational conservation of non-structural proteins, the in silico prediction of the Nsp of alpha, beta, gamma, and delta coronaviruses was carried out by using the fully automated SWISS MODEL. Expasy and the DeepView program (Swiss Pdb-Viewer), and the SWISS-MODEL repository via database to the up-to-date collection of annotated 3D protein models generated by automated homology models for relevant model organisms and experimental structure information for all sequences in UniProtKB were used [9]. 

### 2.3. Evolutionary Relationships of Nsp Proteins in Coronavirus Families

The identification of conserved structural domains and non-structural proteins Nsp3, Nsp5, Nsp6, Nsp9, Nsp12 (RdRp), Nsp13, Nsp14 was delimited by CDD (https://www.ncbi.nlm.nih.gov/Structure/cdd/wrpsb.cgi; accessed on 19 February 2023) [10]. For the construction of the NSP Cluster, the selected proteins and the alignment of conserved amino acids were included, and the consensus sequence was obtained by ClustalW by using the Sequence Alignment Editor BioEdit program v 7.0.5.2 (https://bioedit.software.informer.com/) (accessed on 19 February 2023). 

Evolutionary analyses were performed in MEGA11 [11], and the Minimum Evolution method was used, with a boopstrap of 1000 replicates. Evolutionary distances were calculated by using the Poisson correction method based on the number of amino acid substitutions per site. The dendrogram was constructed by using the Close-Neighbor-Interchange (CNI) algorithm. 

## 3. Results

### 3.1. Annotation of the Alpha, Beta, Gamma, and Delta Coronaviruses Genomes

In this study, we analyzed representative genomes of alpha, beta, gamma, and delta coronavirus. For alphacoronaviruses, the genome of coronaviruses 229E (HcoVs-229E) and NL63 (HcoVs-NL63) with a size of 27,317 nt (NC_002645) and 27,553 nt (NC_005831), respectively, were explored. Betacoronaviruses OC43, HKU1, MERS-CoV and MERS-CoV-Eng have a genome size of 30,741 nt, 29,926 nt, 30,119 nt, and 30,111 nt, respectively. Within this group are the coronaviruses related to severe acute respiratory syndrome SARS-CoV-Tor2, SARS-CoV, and SARS-CoV-2 with an RNA genome of approximately 29,000 nt. Finally, within the deltacoronaviruses, the porcine virus HKU15 and the sparrow virus HKU17 with a genomic ssRNA length of ~25,000 nt were selected (Appendix A).

The pangenome of the coronaviruses HCoVs-229E, HCoVs-NL63, HCoVs-OC43, HCoVs-HKU1, MERS-CoV, SARS-CoV, SARS-CoV-Tor2, AcCoV-Duck, AcCoV-Turkey, HKU15, and HKU17 indicated 121 coding genes, of which 82 and 39 belong to the homologous and unique genes (singleton) of the CoV variant. The total number of families considered for the pangenome was 69 families, 30 of which were homologous and 39 of which were unique (Appendix A).

The comparison of the thirteen selected coronavirus genomes demonstrated a total of 69 protein families and 57 functions associated with the structural domains found in each genome. The SARS-CoV-2 Betacoronavirus presented greater homology with the SARS-CoV and SARS-CoV-Tor2 genomes (Appendix A). Within the pangenome, functions associated with spike glycoprotein, envelope protein, hemagglutinin esterase, hemagglutinin esterase glycoprotein, M membrane glycoprotein, nucleocapsid phosphoprotein, replicase polyprotein, small envelope protein, spike surface glycoprotein, and related functions were found with non-structural proteins (Nsp) (Table 1). The analysis of the five Betacoronavirus genomes identified 53 coding genes, 40 of which were homologous and 13 of which were specific genes. The protein families were 29 (16 homologues and 13 specific genes at each genome) (Appendix A). Even though the SARS and MERS Coronaviruses belong to the B lineage, a divergence was observed in the protein families by genome. Thus, SARS-CoV-2 presents only one homologous family with MERS-CoV and MERS-CoV-Eng. This section may be divided by subheadings. It should provide a concise and precise description of the experimental results and their interpretation, as well as the experimental conclusions that can be drawn.

### 3.2. Structure of the SARS-CoV-2 Genome

SARS-CoV-2 presented a positive single-stranded RNA genome of 29,903 nt in length, consisting of ORF1ab with a size of 7096 aa (YP_009724389.1) encoding non-structural proteins Nsp flagged 1 to 16 that appeared during their intrahost reproductive cycle. It was found flanking the 5′UTR end and the ORF that codes for structural proteins is located at the 3’UTR end. The structure of the SARS-CoV-2 genome (NC_045512) identified the following regions: (A) untranslated region at the 5′ end (5′UTR) that in this group of coronaviruses has been described to be potentially translatable and within this, short ORFs have been described that encode peptides of approximately 3 to 11 amino acids (stem-loop 1,2); (B) the untranslated region at the 3′ end (3′UTR ) that has a GGAAGAGC sequence of a 73 to 80 nt octamer upstream of the poly(A) tail (s2m) and a 3′-terminal poly(A) tail; (C) the 1ab ORF, which codes for non-structural proteins (Nsp1 to 16) proteins involved in the proteolytic processing of polyprotein 1, replication of the virus genome, and RNA synthesis (transcription). Polyprotein 1 results from the translation of ORFs 1a and 1ab and from ribosomal displacement induced by the pseudoknot stem-loop 1,2 with the sequence UUUAAAC; (D) ORF 1a, which encodes mainly for structural proteins involved in the formation of the virion, such as the structural spike glycoprotein (S), the envelope protein; (E) the membrane glycoprotein (M) and the protein of the nucleocapsid (N) (Appendix A, Figure 1). 

### 3.3. Identification of Nsp Domaizns in the Coronavirus Genome

Each virus genetic variant was analyzed for coronavirus polyprotein (ORF1ab), which contains overlapping open reading frames encoding PP1ab (ORF1ab) polyproteins. These polyproteins are cleaved to produce 16 non-structural proteins, Nsp1-16. The production of the longer (ORF1ab) or shorter (ORF1ab) protein depends on a ribosomal frameshift event. An analysis of conserved domains from protein families indicated a homology between the CDS of the non-structural proteins Nsp2, Nsp3, Nsp4, Nsp5 called major protease (Mpro), Nsp6, Nsp8, Nsp10, the domain of RNA-dependent RNA polymerase (RdRp), known as Nsp12, helicase domain, zinc-binding, and stem DC of Nsp13, Nsp14, N-terminal domain of Nsp15, and papain-like protein (PLPro) (Figure 2, Table 2).

In alpha and beta coronaviruses, conserved regions with the gamma and delta coronavirus genera were identified, such as the enzymes Nsp7 (cd21826: alphaCoV_Nsp7), Nsp9 (cd21897: alphaCoV_Nsp9), Nsp13 helicase (cd21409: 1B_cv_Nsp13-like), the specific uridylate endoribonuclease nidovir to (NendoU), Nsp15 and related proteins (cd21161: NendoU_cv_Nsp15-like), the middle domain of Nsp15 (cd21167: M_alpha_beta_cv_Nsp15-like), and the X or Mac1 domain of Nsp3 and related macrodomains (cd21557: Macro_X_Nsp3-like) (Table 2).

Among the specific domains of protein families conserved for SARS-CoV-2 of lineage B, we found: the C-terminal domain of Nsp1 (cd22662: SARS-CoV-like_Nsp1_C), CDS of the enzyme 2′-O-methyltransferase ( pfam06460: CoV_Methyltr_2), Nsp3 nucleic acid-binding domain (cd21822: SARS-CoV-like_Nsp3_NAB), Nsp3-specific marker (cd21814: SARS-CoV-like_Nsp3_betaSM), the SARS C-terminal unique domain (SUD) of the Nsp3 (cd21525: SUD_C_SARS-CoV_Nsp3), Nsp3 single-stranded poly(A) binding domain (cl13138: SUD-M Superfamily), and the CDS of the macrodomain superfamily (cl00019: Macro_SF Superfamily) (Table 2).

In the case of the human coronavirus NL63 (HCoV-NL63) of the alphacoronavirus genus of lineage A, specific domains were detected, such as the C-terminal region of the Nsp2 replicase (pfam19212: CoV_NSP2_C), the CDS of the papain-like protease (PLPro) from Nsp3 (cl40457: CoV_PLPro Superfamily). The C-terminal domain of the Nsp2 replicase (pfam19212: CoV_NSP2_C) was found in both alpha and betacoronaviruses (Graham et al., 2005). This domain was found in two copies in some Nsp2 viral proteins. The papain-like protease (PLPro) domain cl40457: CoV_PLPro Superfamily is a characteristic region of Nsp3 described in alpha-, beta-, gamma-, and deltacoronaviruses [12].

In the AcCoV-Turkey gammacoronavirus (YP_001941164), the specific domains were for the pneumococcal surface protein PspC (cl41463: PspC_subgroup_2 Superfamily) and the ubiquitin-like domain (Ubl) located at the N-terminus of Nsp3 (cl28922: Ubl1_cv_Nsp3_N-li Superfamily). The pneumococcal surface protein, PspC, is a repetitive and highly variable protein, recognized by a conserved N-terminal domain that is covalently anchored after cleavage at the C-terminal LPXTG site. There is a genetic variant that has a choline-binding repeat in the C-terminal region and is known as choline-binding protein A [13]. The cl28922: Ubl1_cv_Nsp3_N-like Superfamily is a ubiquitin-like (Ubl) domain (Ubl1) found at the N-terminus of coronavirus Nsp3, a multi-domain multifunctional protein that is an essential component of the replication/transcription complex (RTC). The functions of Ubl1 in CoVs are related to single-stranded RNA (ssRNA) binding and interaction with the nucleocapsid (N) protein. SARS-CoV Ubl1 has been shown to bind to ssRNA with AUA patterns from the 5′-UTR of the SARS-CoV genome. This region corresponds to one of the two Ubl domains in Nsp3 (Sapir et al., 2000; Goehring et al., 2003). Finally, the ORF of the 1ab polyprotein of the deltacoronavirus HKU17 (AWV67106.1) lacked specific CDS but shared a greater number of domains with the alphacoronavirus genus (Table 2).

### 3.4. In Silico Tertiary Structures of Nsp from Alpha, Beta, Gamma, and Delta Coronaviruses

The conformation of the tertiary structure of Nsp3 of the alphacoronaviruses, HCoVs-229E and HCoVs-NL63, demonstrated homology with the papain-like protease (PLpro) reference 6I5t.1 that presents a right hand composed of three domains (the thumb, palm, and fingers) (Figure 3A, Appendix A). The secondary structure arrangement of the catalytic domain has seven α-helices (six thumb domains, one finger domain), thirteen total β chains (two thumb domain, four finger domain β chains, and seven palm domain β chains), and one helix (η) in the thumb domain and another helix (η) in the finger domain. The thumb domain is predominantly α-helical and contains six α-helices and two short β-strands that are arranged in a β-hairpin. The thumb domain interacts with the palm domain via the β-hairpin. Although the finger domain is made up of three long and one short β chains, the palm is made up of seven β chains. The Nsp3 tertiary structure of Betacoronaviruses showed a differential pattern between HCoVs (OC43, HKU1), MERS-CoV (Figure 3B, Appendix A), and SARS-CoV. 

The first two groups of coronaviruses were modeled by using the 3D reference 5v69.1, which corresponds to the molecular basis of PLpro MERS-CoV inhibition by ubiquitination (UbV). While promoting more favorable hydrophobic interactions with the enzyme in relation to ubiquitination, these extensive interactions at the PLpro-Ub interface are characteristic of each group of coronaviruses tested.

The structural conformation of the Nsp3 of the gammacoronaviruses was compared with the 3D reference of the MERS (5v69.1), a structure of differential arrangement was observed with a distant identity of 26.33% (AcCoV-Duck) and 23.59% (AcCoV-Turkey) (Figure 3C/Appendix A). Regarding the Nsp3 of the deltacoronaviruses, two 3D references were used to model each accession, which were HKU15 (7rqg.2) and HKU17 (6In0.1) with 20% and 96% identity of the PLpro domain (Figure 3D, Appendix A). Reference 7rqg.2 comprises the Y3 region of the Nsp3 enzyme, this domain showed low identity to the porcine coronavirus HKU15 (QDH76194.1). In the case of the Sparrow coronavirus HKU17 (AWV67106.1), 6In0.1 was used as reference, which spans three Nsp3 domains. The crystal structure of multiple functional domains of porcine delta-coronavirus (6In0.1) includes the Macro domain, ubiquitin-like domain 2 (Ubl2) and the papain-like protease catalytic domain (Plpro). In the asymmetric unit, two of the subunits form the N-terminal to C-terminal homodimer with an interaction interface between Macro and Plpro [14].

The analysis of tertiary structures of Nsp5 of the alphacoronaviruses (HCoVs-229E, HcoVs-NL63) demonstrated an identity of 69% and the betacoronaviruses an identity ˂50%, with the reference 6w81.1 corresponding to the dimerization structure of the main protease in coronaviruses. The conformational folding of non-structural protein 5 from these two groups was similar to the reference 3D (Figure 4A,B; Appendix A). The predicted structures of Nsp5 of gamma (AcCoV-Duck, AcCoV-Turkey) and deltacoronavirus (HKU15 and HKU17) were approximately 40% identical, using the three-dimensional structure of reference 4xFq.1, which corresponds to the crystal structure of PEDV 3CLpro of two asymmetric molecules with domain I, II, and III. The first two domains exhibit an antiparallel β-barrel structure, which is like the 3CLpros of other coronaviruses [15]. Domain III consists of five α-helixes, and it is connected to domain II by a loop. In the 3Clpro structure, the substrate binding site is located in a cleft between domains I and II (Figure 4C,D, Appendix A).

A comparison of the tertiary conformation of Nsp6 demonstrated low identity values in all tested coronavirus genera (Appendix A), due to the partial region of 300 amino acid residues from the C-terminus. The reference ID was 2k7x.1 of the 3D structure of the C-terminal domain of the major protease of SARS-CoV. In the alpha and delta coronavirus groups, the monomer conformation of the C-terminal domain of Nsp6 was obtained, while in the beta and gamma coronavirus accessions, they exhibited the folding of a dimer (Appendix A). It has been described that the conformational structure of the C-terminal domain of coronavirus Nsp6-Mpro can adopt monomer and dimer structures. The Mpro-C monomer maintains the same fold as in the structure derived from crystallography. Conversely, the Mpro-C dimer has a new structure characterized by domain exchange that provides the base structure for the stability of the dimer [16]. 

The results of the comparative analysis of coronavirus Nsp9 indicated a high conservation in the folding of the tertiary structure in three groups of coronaviruses except for alphacoronavirus. Despite low identity (35 to 50%) of Nsp9 in genera gamma, delta and in some betacoronavirus accessions, conformational homology was obtained with reference 1qz8.1 and 6w4b.1, respectively (Figure 5, Appendix A). 1qz8.1 indicates the crystal structure of a SARS coronavirus Nsp9 dimer, where each monomer contains seven strands and one helix arranged in a single compact barrel-shaped domain flanked by the C-terminal helix. The latter forms an angle of 45° with the barrel axis and has a high content of hydrophobic residues ID 6w4b.1 points to the SARS CoV-2 Nsp9 structure, which forms a dimer at the N-terminus of the β-strand and C-terminus of the α-helix contributing to the dimer interface [17].

In general, Nsp12 has a tertiary structure that is conserved among coronaviruses. Subtle conformational variation was observed among alphacoronaviruses. Reference model 6xez.1.A indicates the structural basis for helicase–polymerase coupling in the SARS-CoV-2 replication–transcription complex, illustrating the RNA-dependent RNA polymerase (RdRp) interaction of the SARS-CoV-2 as a holoenzyme with Nsp7/Nsp8/Nsp12 containing an RNA template in conjunction with viral helicase (Figure 6, Appendix A). RNA-dependent RNA polymerase (RdRp), known as Nsp12, is a central component of the viral RNA replication and transcription complex. The Nsp12 complex with the Nsp7 and Nsp8 cofactors has catalytic activity and is a minimal central component for virus RNA replication. The structure of the SARS-CoV-2 RdRp complex consists of an Nsp12 core catalytic unit and an Nsp7–Nsp8 heterodimer. The N-terminal portion of Nsp12 contains a β-hairpin and a Nidovirus-specific extension domain (NiRAN). The C-terminal catalytic domain of Nsp12 connects to NiRAN via an interface subdomain [18]. The RdRp active catalytic site consists of seven conserved motifs. Motifs A, B, C, and D are from the palm subdomain and in motif C it forms the active catalytic center. F and G motifs are found within the finger subdomain that interact with RNA towards the active site [19]. 

For the in silico prediction of the tertiary structure of coronavirus Nsp13, the ID: 5wwp.2, which corresponds to the crystal structure of the helicase of the Middle East Respiratory Syndrome coronavirus (MERS-CoV), was used as a reference. The putative helicase tertiary structures in this study were similar in conformation; mainly, MERS-CoV demonstrated full 3D (Figure 7, Appendix A). The general structure of Nsp13 consists of five different domains arranged in a triangular shape. Three domains that form the base of this pyramid are connected via a stem domain to the N-terminal zinc-binding domain (ZBD) that forms the apex of the pyramid. The structural assembly is very well organized to allow all five domains of Nsp13 to participate in helicase activity in a coordinated manner [20].

The tertiary structure of Nsp14 indicated a conserved conformation of this bifunctional protein among the alpha, beta, gamma, and delta coronavirus genera. The reference 3D was 5nfy.2 of the canonical structure of the Nsp14 complex with its cofactor Nsp10 from SARS-CoV (Figure 8, Appendix A). Nsp14 is a bifunctional protein carrying guanine N7-methyltransferase (MTase) and 3′-5′ exoribonuclease (ExoN) activities involved in RNA virus methylation which is important for viral mRNA stability by evading its degradation from the host immune response and the proofreading activity of Nsp14 has been suggested to be responsible for the evolution and maintenance of large coronavirus genomes. Nsp14 contains four structural regions: the Nsp10 binding site, an ExoN domain, a flexible hinge region, and a C-terminal N7-MTase. It has been suggested that the binding of the cofactor, Nsp10, stabilizes the active site to the correct conformation for catalysis [21].

### 3.5. Comparative Analysis of the Tertiary Structural Conformation of Nsp in SARS-CoV 

The tertiary structure of Nsp3 generated in silico of the betacoronavirus SARS demonstrated a high similarity in the conformation of the domain papain-like protease PLPro and an identity greater than 80%. Thus, SARS-CoV (APO40578.1), SARS-CoV-2 (YP_009725299.1) and SARS-CoV-Tor (NP_828862.2) obtained 80.52%, 99.67%, and 82.41% identity (Figure 9A/Appendix A). The reference ID was 7d6h.1 of the PLPro-like consisting of four subdomains (N-terminal ubiquitin-like domain (Ubl2), thumb, zinc finger, and palm subdomain) and follows a “thumb–palm–finger” architecture that form the catalytic part. The Ubl2 subdomain consists of a α-helix and five β chains. The thumb consists of six α propellers and a β fork. The palm subdomain has six β chains and contains the Cys, His, and Asp residues that make up a catalytic triad of its interface with the thumb subdomain. The finger subdomain consists of six β strands and two α-helices. Within the loops and chains β reside four conserved cysteine residues (Cys189, Cys192, Cys224, Cys226), which form a zinc finger and coordinate a zinc ion. The PL2pro domain exhibits two different binding sites for cleavage of ubiquitin molecules or ISG15; both binding sites differ in substrate specificity and activities [22].

For analysis of the betacoronavirus Nsp5, two reference points were used, which were 7dpp.1 and 2a5i.1. The conformation of the structures was like the reference with an identity of 95% (SARS-CoV) and 100% (SARS-CoV-2 and SARS-CoV-Tor), Figure 9B/Appendix A. Nsp5-3CLpro 3D structures showed great similarity, especially in the area around the active site, and small changes were found mainly in domain III between SARS-CoV and SARS-CoV-2. Domains I and II showed a higher degree of conservation compared to domain III between different genera of coronavirus. The reference ID corresponds to a 33 kDa cysteine protease, known as a major protease (Mpro) or 3C-like protease (3CLpro), the structure of SARS-CoV-2 Mpro functions as an active homodimer with an approximate C2 symmetry, like other CoV Mpros. Each protomer has three domains (I, II, and III). Domains I and II adopt a six-stranded antiparallel β barrel fold. A deep indentation between domains I and II forms the substrate binding site, where the N-terminal finger (domain I) plays an important role in catalysis. The C-terminal III globular domain comprises five antiparallel helices and participates in dimerization through an intermolecular salt bridge interaction. Domains II and III are connected by a loop exhibiting two distinct conformations, a widespread conformation, or the formation of a short helix [23].

The folding of a partial region of the C-terminal domain of Nsp6 demonstrates a conformation similarity, but low similarity values with respect to the reference 3D 7nbu.1. In SARS-CoV and SARS-CoV-2, we observed the main conformation of α-helices and in SARS-CoV-Tor, β-helices in the C-terminal of the membrane glycoprotein (Figure 9C/ Appendix A).

The results of the in silico prediction of the Nsp9 of the betacoronavirus genus indicated a conservation in the conformation of the Nsp9 enzyme in SARS-CoV and SARS-CoV-Tor particularly, with an identity of 98 and 100%, respectively (Figure 9D/Appendix A). ID Iqz8.1 is a single-stranded RNA-binding protein involved in virus virulence. SARS-CoV-2 Nsp9 shares a 97% sequence identity with the SARS-CoV orthologue. The structure of Nsp9 consists of a series of external extended loops projecting from the β barrel core (closed conformation of 6 strands binding to RNA). Nsp9 forms a dimer with the N-terminal β strand and the α1 C-terminal helix contributing to the dimer interface. The C-terminal portion of the helix forms funnel-shaped hydrophobic cavities. 

The tertiary structure of RNA-dependent RNA polymerase (RdRp) indicated a high conservation and identity greater than 95% in SARS-CoV, SARS-CoV-2, and SARS-CoV-Tor. Reference ID 7bv1.1 corresponds to the Nsp12 holoenzyme complex with cofactors Nsp7 and Nsp8 with catalytic activity in SARS-CoV-2 (Figure 9E, Appendix A). The Nsp12-RdRp domain displays the canonical configuration in cup form, with the finger subdomain forming a closed structure with the thumb subdomain. The closed conformation of Nsp12 is held stable by the Nsp7 and Nsp8 heterodimer [24]. The Nsp7–Nsp8 heterodimer of SARS-CoV-2 shows a conserved structural similarity to that of SARS-CoV [24].

The structure of the Nsp13 helicase demonstrated high similarity to SARS-CoV and SARS-CoV-2 with a conformational change in SARS-CoV-Tor. Reference 5wwp.2. corresponds to the crystal structure of MERS-CoV Nsp13 with the conformation of the five structural domains for helicase activity. Within this coronavirus group, an identity greater than 71% was determined (Figure 9F, Appendix A).

The coronaviruses SARS-CoV, SARS-CoV-2, and SARS-Tor had a similar conformation of Nsp14; however, in comparison with the reference heterodimer 5nfy.2 (Nsp14 complex with its SARS-CoV cofactor Nsp10), an identity 52% was observed and it was observed absence of the cofactor (Figure 9G, Appendix A). The reference 3D was 5nfy.2, which corresponds to the canonical structure of the bifunctional protein carrying guanine N7-methyltransferase (MTase) and 3′-5′ exoribonuclease (ExoN) activities.

### 3.6. Evolutionary Relationships of Non-Structural Proteins (Nsp) of Different Coronavirus Genus

In this research, we focused on determining the evolutionary relationships of non-structural proteins in human pathogenic coronaviruses. For this, 115, 109, 95, 95, 137, 84, and 115 accessions of Nsp3 (Appendix A), Nsp5 (Appendix A), Nsp6 (Appendix A), Nsp9 (Appendix A), Nsp12 (RDPD, Appendix A), Nsp13 (Appendix A), and Nsp14 (Appendix A) of the alpha, beta, gamma, and delta coronavirus genera were selected by using the NCBI database. With this set of accessions, the structural conserved domain of each Nsp that oscillated in length was determined.

The CoV_PLPro (alphacoronavirus papain-like protease, Nsp3) was from 289 to 313 aa, the Mpro (Main protease (Mpro Nsp5), Nsp6 various from 290 to 307, the Nsp 9 was from 108 to 113 aa, the RdRp domain (Nsp12) demonstrated a highly conserved domain from 924 to 931 aa. The helicase Nsp13 and Nsp14 were delimited a CDD of 340 to 343 and 508 to 519 aa, respectively (Appendix A). With these accessions, the consensus sequence of each non-structural protein of the different groups of coronaviruses was obtained, which presented a length of approximately 2000 aa residues (Appendix A). With the consensus sequence obtained from the genetic dataset of each genus of coronavirus, the evolutionary relationships of the non-structural proteins (Nsp) of the entities with pathogenicity in humans were established (Figure 10).

The results of analysis of evolutionary relationships indicated a classification pattern in each non-structural protein (Nsp3, -5, -6, -9, Rdpd-12, -13 and -14) according to gender category. Betacoronaviruses were the entities with the greatest genetic diversity demonstrated mainly by Nsp9, Rdpd (Nsp12), and Nsp5 (Figure 10). The Nsp cluster that was constructed indicated that it could be used as a lineage marker to delimit an optimal classification for betacoronaviruses that are the most virulent pathogens in humans and that are currently evolving rapidly, which is reflected in the emergence of new coronavirus variants. SARS-CoV-2 presented a positive single-stranded RNA genome of 29,903 nt in length, consisting of ORF1ab with a size of 7096 aa (YP_009724389.1) encoding non-structural proteins Nsp flagged 1 to 16 that appear during their intrahost reproductive cycle. It was found flanking the 5′UTR end and the ORF that codes for structural proteins is located at the 3′UTR end.

## 4. Discussion

### 4.1. Functional Annotation and Identification of Non-Structural Protein (Nsp) Domains in the Coronavirus Genome

The results indicated a marked evolutionary conservation of Nsp that allowed us to identify and classify highly pathogenic human coronaviruses of the alpha, beta, gamma, and delta groups. However, an analysis of characteristic domains of conserved protein families and specific regions for enzyme binding responsible for viral transcription, replication, proteolytic processing, suppression of host immune responses, and host gene expression demonstrated that some of the non-structural proteins (Nsp) are divergent in coronavirus genera. Some have coevolved with other regions as part of a specific genetic unit in some coronaviruses (CoV). CoVs have lower mutation rates than other respiratory viruses. However, many infections cause an expansion in viral genetic variability causing epidemics to result in increasing genomic diversity among hosts [25]. These types of evolutionary studies provide a window of opportunity to employ these non-structural proteins (Nsp) as targets of viral therapies as is being scientifically advanced with RNA-dependent RNA polymerase. Likewise, the identification of evolutionary markers allows us to follow their evolution trajectory and know the origin of the lineage of genetic divergence impacting on its viral classification and the search for biomedical technology for its application in the field of health. Authors should discuss the results and how they can be interpreted from the perspective of previous studies and of the working hypotheses. The findings and their implications should be discussed in the broadest context possible. Future research directions may also be highlighted.

### 4.2. Specific Domains of Nsp Can Be Used as Coronavirus Identification Markers

The N-terminal and C-terminal domains of the non-structural protein 1 (Nsp1) of betacoronaviruses, cd21796: SARS-CoV-like_Nsp1_N and cd22662: SARS-CoV-like_Nsp1_C, respectively, were identified. Nsp1 alphaCoV and betaCoV share structural similarity, do not show significant sequence similarity, and can be considered taxonomic group-specific markers at the genus level. Despite low sequence similarity, alphaCoV and betaCoV Nsp1s exhibit remarkably similar biological functions and are involved in the regulation of viral and host gene expression [26,27,28].

Within the unique domains found in B-lineage betacoranaviruses, we found the nucleic acid-binding domain (NAB) of non-structural protein 3 (Nsp3), cd21822: SARS-CoV-like_Nsp3_NAB, is a cytoplasmic domain located between papain-like protease (PLPro) and CoV Nsp3 betacoronavirus-specific marker (betaSM) domains [29,30]. cd21814: SARS-CoV-like_Nsp3_betaSM representing the betacoronavirus-specific marker (betaSM) or group-2-specific marker (G2M) of the Nsp3 protein. betaSM/G2M is located at the C-end of the nucleic acid-binding (NAB) domain. In gammacoronaviruses, we found a characteristic domain of this group (ammas); but this region is absent in the Nsp3 of alpha and deltacoronavirus. The betaSM/G2M domain of Nsp3 (composed of 385 amino acids) of SARS-CoV can function as a replication/transcription scaffold, with interactions with Nsp5, Nsp12, Nsp13, Nsp14, and Nsp16 [31,32]. Additionally, in SARS-CoV-2, we found the unique domain of SARS C-terminal (SUD) of non-structural protein 3 (Nsp3), cd21525: SUD_C_SARS-CoV_Nsp3, which consists of three globular domains separated by short peptide segments: SUD-N, SUD-M and SUD-C. In the alpha, beta, and deltacoronavirus groups, two domains of Nsp7 and 15 were identified, which were cd21827: betaCoV_Nsp7 is involved in the processing of four proteins Nsp7, Nsp8, Nsp9, and Nsp10 by protease M (Mpro), which form functional complexes with central CoV enzymes and stimulate replication. It has been shown that a complex of Nsp7 with Nsp8 activates and confers processivity to the synthesis activity of RNA-dependent RNA polymerase (RdRp). Nsp7 has a 4-helix beam conformation that is strongly affected by its interaction with Nsp8. Nsp7 in SARS-CoV forms a hexadecameric complex with Nsp8 that adopts a hollow cylindrical structure with central channel and positive electrostatic properties in the cylinder; this Nsp7/Nsp8 complex functions as a non-canonical RNA polymerase capable of synthesizing an RNA template [33,34,35], and in Nsp15, the nidoviral-specific uridylate endoribonuclease domain (NendoU), cd21161: NendoU_cv_Nsp15-like, involved in the process of viral replication and evasion of the host immune system was identified. Severe acute respiratory syndrome coronavirus (SARS-CoV) Nsp15, human coronavirus 229E (HCoV229E), and murine hepatitis virus (MHV) form a functional hexamer, while porcine deltacoronavirus (PDCoV) exists as dimer and monomer [36,37].

### 4.3. Changes in 3D-Structure of Nsp Suggest Adaptation of Coronaviruses during Their Viral Infection Mechanism

The in silico analysis of tertiary structures of the Nsp identified in the genome of coronaviruses of the genus alpha, beta, gamma, and delta detected the conservation of the tertiary structure, despite its low identity with the reference model, which suggests that these proteins have a high flexibility capacity to modify their folding for the recognition of receptors or signal molecules during their viral infection mechanism. This behavior may be a window of opportunity to promote the use of inhibitors in the antiviral response.

A similarity analysis of different Nsp domains demonstrated a higher identity in Nsp9 followed by Nsp5, Nsp3, and Nsp6. Some of these non-structural proteins presented specific domains in each phylogenetic group, as Nsp3 domains are present in all coronaviruses, while others only exist in specific coronaviruses. The multifunctionality of this protein coupled with the inconsistent domain nomenclature makes it difficult to fully understand. So far, no complete structure of Nsp3 has been solved and structural models are available for only six of its seventeen domains. These include ubiquitin-like domains (Ubl1 and Ubl2), macrodomain 1 (Mac1), papain-like protease (PL2pro), and nucleic acid-binding domain (NAB), as well as the Y3 domain. 

The Nsp9 protein demonstrated a conformation in the structure of this conserved enzyme between SARS variants (CoV, -CoV-Tor, -CoV-2), indicating that this enzyme may be a target to deter diseases caused by this virus. In addition, structural information suggests that compounds that disrupt the Nsp9 dimer interface may be developed as drugs against coronavirus diseases [38,39].

Among the non-structural proteins, the RNA-dependent RNA polymerase (Rpd, Nsp12) stands out, forming a replicase complex with Nsp7 and Nsp8 that are indispensable for the replication and transcription of the viral RNA genome. In this study, the conservation of Nsp12 in SARS-CoV was observed with variation in Nsp12 conformation in alphacoronavirus and MERS-CoVs accessions. Enzymes that are vital to the viral replication cycle are suitable antiviral drug targets because they differ from host proteins. Among viral enzymes, RdRp is the primary target of many existing nucleotide drugs, such as remdesivir favipiravir, ribavirin, galidesivir, and several nucleotide analog drugs.

The Nsp13 protein of coronavirus presented a similarity in tertiary structure. The structure of Nsp13 with the SARS-CoV-2 replication and transcription complex consists of Nsp7, Nsp8, and Nsp12 with potential implications for helicase activity and regulation. Due to its essential role in viral replication and conservation in all CoV species, Nsp13 is an important target for antiviral drug development [40,41].

For its part, Nsp14 demonstrated a tertiary conformation conserved in all coronavirus groups. This protein plays an important role during genome replication and the protective activity of viral mRNA makes it an important target for antiviral drug development. It has been suggested that the corrective function of Nsp14 acts as a barrier to the development of antivirals against SARS-CoV-2.

Likewise, the genus of Betacoronavirus stands out with a marked pattern between the viral variants that cause severe acute respiratory syndrome (SARS) and Middle East respiratory syndrome (MERS) that used different 3D reference structures. Among the accessions of SARS, the topology presented by the representative of SARS-CoV-2 with respect to some Nsp was distanced.

### 4.4. Possible Drug Binding Sites of Non-Structural Proteins (Nsp) in Coronaviruses

The tertiary structures of Nsp indicate target sites of binding to complex molecules, bioactive molecules, or drugs, depending on their architecture and conformational folding. In Nsp3, it has been described that the catalytic site presents a “thumb–palm–finger” architecture, the palm subdomain contains a catalytic triad of amino acid residues (Cys111, His272, and Asp286) at its interface with the thumb subdomain. At the active site is a blocking loop 2 (BL2) involved in regulating substrate binding and changing its conformation when bound to a substrate or inhibitor. Within the catalytic center is the PL2pro domain that exhibits two different binding sites for the cleavage of ubiquitin molecules or ISG15 (ubiquitin-like enigma). Both binding sites differ in the specificity and activities of the substrate, which can be used for targets in viral therapies [22,42,43,44].

Nsp 5 is a cysteine protease involved in the formation of complex for virus genome replication [45,46]. In SARS-CoV, this protein is targeted by the design of inhibitors with an effective response against the disease caused by the coronavirus, for example, the Carmofur and Ebselen [47,48]. There are a number of inhibitors that have been analyzed with the crystal structure of Nsp5 SARS-CoV-2, which highlights this protein as a potential target for the development of antiviral therapies due to its indispensable participation during viral replication [25].

Of the 16 Nsp of coronaviruses, most interact to form the complex of replication and functions related to cellular processes. So far, the precise stoichiometry of the replicase complex is unknown. In this study, we analyzed only the conformation of a partial region of the C-terminal domain of Nsp6, where the α helices and β helices of the membrane glycoprotein were located. The Mpro-C monomer maintains the same fold as in the structure derived from crystallography, where the presence of multiple residues of phenylalanine in the region of the outer membrane of Nsp6 could favor the affinity of this region and the membrane to the endoplasmic reticulum, inducing a more stable binding of the protein to this cellular organelle. It has been shown that this binding may favor coronavirus infection by compromising the ability of autophagosomes to deliver viral components to lysosomes for degradation [49]. Therefore, by blocking this region rich in phenylalanine residues, there is a possibility of deficiencies in the process of autophagy and inflammatory response of the host immune system [6,25,50,51].

During the viral cycle, the proteins that interact for the formation of the replication complex are Nsp7, Nsp8, Nsp12, and Nsp13. The function of Nsp8 is to catalyze the synthesis of RNA primers for primer-dependent primase. Nsp12 catalyzes viral RNA synthesis with the help of Nsp7-Nsp8 and is a key target for nucleotide analog antiviral inhibitors such as remdesivir, which has been reported to effectively inhibit SARS-CoV-2 proliferation [52,53]. While Nsp13 exhibits helicase and ATPase activity, unwinding nucleic acid for RNA synthesis by Nsp12. It is important to understand the architecture of the replication complex that is still in the process of architectural elucidation. In this study, it was observed that each Nsp presents catalytic sites for the interaction of molecules and ligands that may affect its participation in the replication complex. Recent studies have been directed to the inhibition of the binding of RNA-dependent RNA polymerase (Nsp12), it has been described that part of the RNA template located in the central channel of Nsp12 during its synthesis; using the drug Remdesivir is covalently incorporated into the primer chain in the first pair of replicated bases to terminate RNA strand elongation, inhibiting viral genome replication [54,55,56].

Nsp9 is a single-stranded RNA-binding viral protein involved in RNA synthesis, essential for coronavirus replication and virulence. NSP9 is composed of seven antiparallel β chains and a single helix α forming a barrel β, the N- and C-terminal regions are more conserved than that of the central core [6,38,57,58]. Nsp9 binds to discrete regions in the 7SL RNA component of the signal recognition particle (SRP) and interferes with protein trafficking to the cell membrane upon infection, interfering with essential host functions and suppressing host immune defenses [56]. Overriding the central site or restricting the interface of the Nsp9 dimer may be a useful strategy for the design of antiviral drugs. 

Nsp14 plays a crucial role in RNA synthesis and in the viral cycle (pathogenicity processes, control of innate immune responses). This protein has a bifunctional domain, the N-terminal exonuclease (ExoN) domain promotes the fidelity of RNA synthesis, while the C-terminal part carries guanosine N7-MTase activity. During the folding of this enzyme, the correct position of the α and β chains at the N-terminus and C-terminus drastically affects the fidelity of replication and protection of the mRNA. These sites may develop as targets for antiviral drugs [6,59,60].

### 4.5. Evolutionary Relationships of Non-Structural Proteins (Nsp) of Different Coronavirus Genus

The evolutionary relationships obtained with non-structural proteins and with the cluster showed that there is a close relationship between the variants of gamma and delta coronaviruses; it has been described as the main enteric pathogens of several mammals (birds, pigs, ferrets, etc.) and the other related group includes the alpha and betacoronaviruses, which are responsible for diseases in different species of mammals, such as respiratory infections in humans and gastroenteritis processes in some animals. Betacoronaviruses are distinguished as the main human pathogen and cause of the COVID-19 pandemic.

Researchers are currently working in the molecular area to identify the source of 2019-nCoV, including potential intermediate animal vectors. A zoonotic reservoir dates to the emergence of SARS and MERS-CoV. SARS CoV, the first highly pathogenic human CoV, emerged in 2002 with transmission from animals to humans [61]. Highly related CoVs have been reported to be identified in bat species [62]. More recent work has shown that several bat CoVs are capable of infecting human cells without the need for intermediate adaptation, as is the case with MERS [58,63]. MERS-CoV likely originated in bats [64,65]. Although camels are endemically infected and contact with camels is frequently reported during primary cases of MERS-CoV [66]. However, in the case of SARS-CoV, it is suggested that it does not require an intermediate reservoir as the virus adapted to infect humans more efficiently [61]. These lessons from SARS and MERS highlight the importance of continuing to advance the knowledge of molecular strategies that allow us to quickly find the source of 2019-nCoV prevent a new outbreak of this coronavirus.

In this research into betacoronavirus, we found subgroups that present a conservation in the genetic unit Nsp, such as human coronaviruses (HCoV-NL63, -229E), Middle East respiratory syndrome coronaviruses (MERS-CoV, -CoV-Eng) and severe acute respiratory syndrome (SARS-CoV, -CoV -Tor, -CoV-2). SARS-CoV-2 appears to be the most recent coronavirus variant defined by the genetic distance found in the Nsp, Nsp3, RdRp, and Nsp14 Cluster.

With the evolutionary trajectory of the Nsp proteins marked with the cluster, it can be suggested as an evolutionary marker that allows us to establish evolutionary relationships and divergent groups mainly between the alpha- and betacoronavirus genera and is positioned as a SARS-CoV-2 lineage marker.

## 5. Conclusions

In this research, the conservation of non-structural proteins of coronavirus variants was highlighted and an Nsp cluster was constructed as a lineage marker for SARS-CoV-2 and evolutionary studies of the alpha- and betacoronavirus genera. Additionally, certain specific changes in the domains of the Nsp were demonstrated, providing an opportunity for the targeted design of drugs that interrupt the replicative cycle and viral infection for the development of biomedical technology for application in the healthcare field.

Despite the advances currently made in the SARS-CoV-2 that caused the COVID-19 pandemic, important challenges remain, such as knowing why SARS coronaviruses have the rapid emergence of new variants because of specific mutations in special regions in the genome and exploring the genetic plasticity presented by betacoronaviruses.

## Figures and Tables

**Figure 1 pathogens-12-01185-f001:**
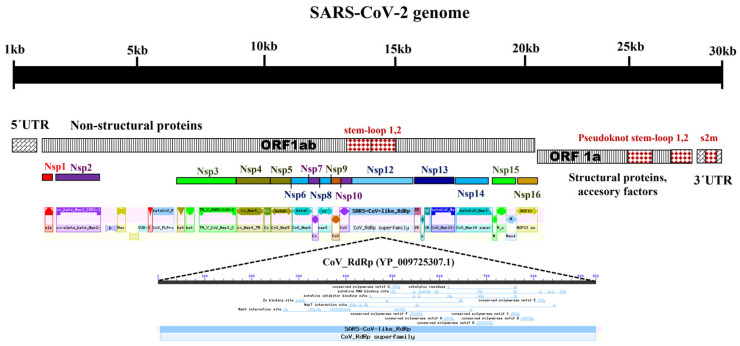
General scheme of the SARS-CoV-2 genome organization. The open reading frame (ORF) 1ab encoding the polyprotein corresponding to the 16 non-structural proteins (Nsp) is illustrated. Among these, the domain of RNA polymerase dependent on coronavirus RNA (CoV-RdRp) known as Nsp12 and the conserved regions of the stem-loop1,2 in the ORF1ab close to 5′UTR stand out. ORF1a represents the coding regions of structural and accessory proteins, as well as Cis elements (pseudoknot stem-loop1,2 and s2m). The genome size of SARS-CoV-2 is approximately 30 Kb in length.

**Figure 2 pathogens-12-01185-f002:**
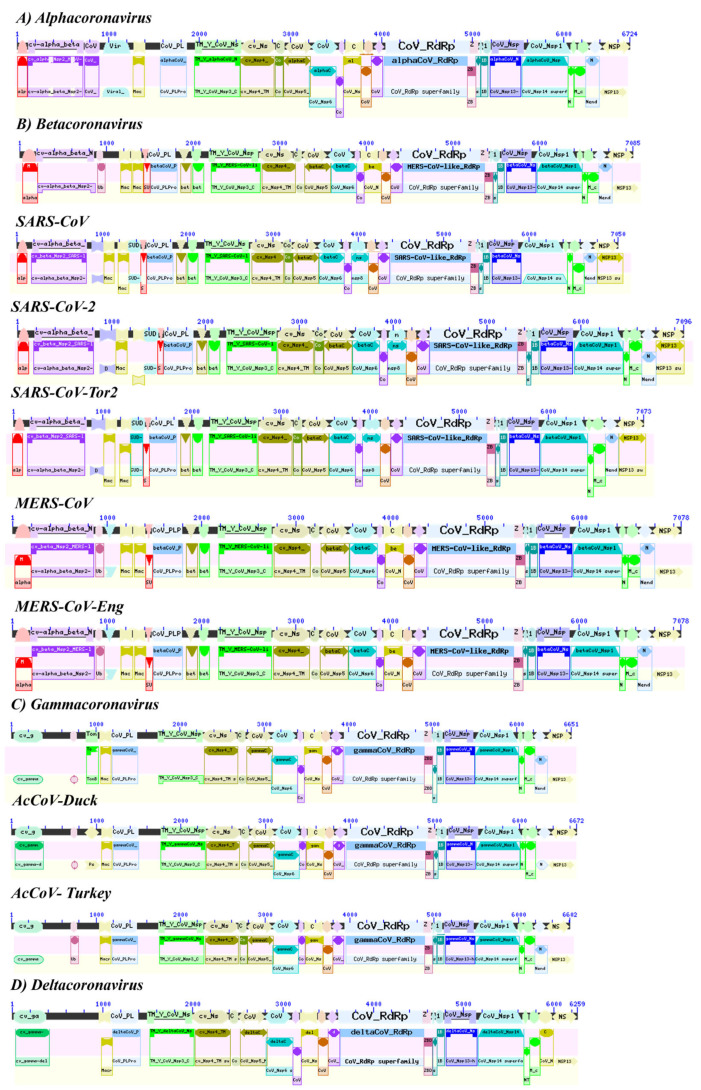
Schematic representation of the identification and location of conserved domains of families of proteins located in the genome of alpha, beta, gamma, and delta coronaviruses using the NCBI Conserved domains tool. The accession numbers of each coronavirus genus were (**A**) Alphacoronavirus (AVA26872.1); (**B**) Betacoronavirus (ATQ39389.1), SARS-CoV (APO40578), SARS-CoV-2 (YP_009724389), SARS-CoV-Tor2 (NP_828849), MERS-CoV (YP_009047202), MERS-CoV-Eng (APO40578.1); (**C**) Gammacoronavirus (QDY92334.1), AcCoV-Duck (YP_009825006), AcCoV-Turkey (YP_00191164.2); (**D**) Deltacoronavirus (AWV67106.1). Protein domain identification was performed with the NCBI CDD software.

**Figure 3 pathogens-12-01185-f003:**
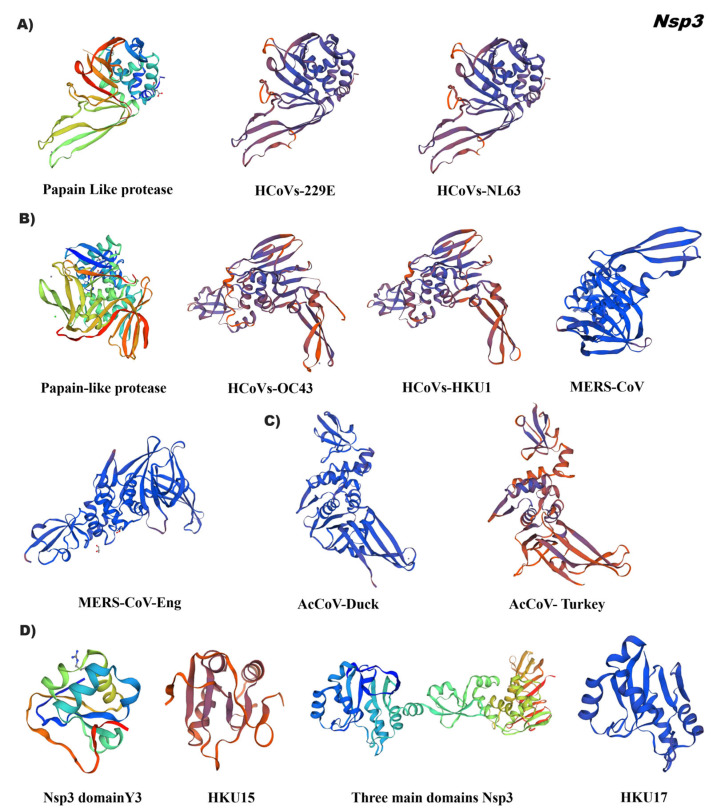
Tertiary structure of papain-like protease (PLpro) Nsp 3 of the coronavirus genus alpha, beta, gamma, and delta. ID corresponding to each virus gene was used as a reference. (**A**) Alphacoronavirus: HCoVs-229E (NP 073549.1), HCoVs-NL63 (YP 003766.2); (**B**) Betacoronavirus: HCoVs-OC43 (YP 009555238.1), HCoVs-HKU1 (YP_173236.1), MERS-CoV (YP_009047215.1), MERS-CoV-Eng (YP_009944294.1); (**C**) Gammacoronavirus: AcCoV-Duck (YP_009825031.1), AcCoV-Turkey (YP_001941176.1); (**D**) Deltacoronavirus: Porcine coronavirus HKU15 (QDH76194.1), Sparrow deltacoronavirus HKU17 (AWV67106.1). Reference ID: 6I5t.1 (Alphacoronavirus), 5v69.1 (Betacoronavirus and Gammacoronavirus), 7rqg.2 and 6In0.1 (Deltacoronavirus). Tertiary structures were made using SWISS MODEL software.

**Figure 4 pathogens-12-01185-f004:**
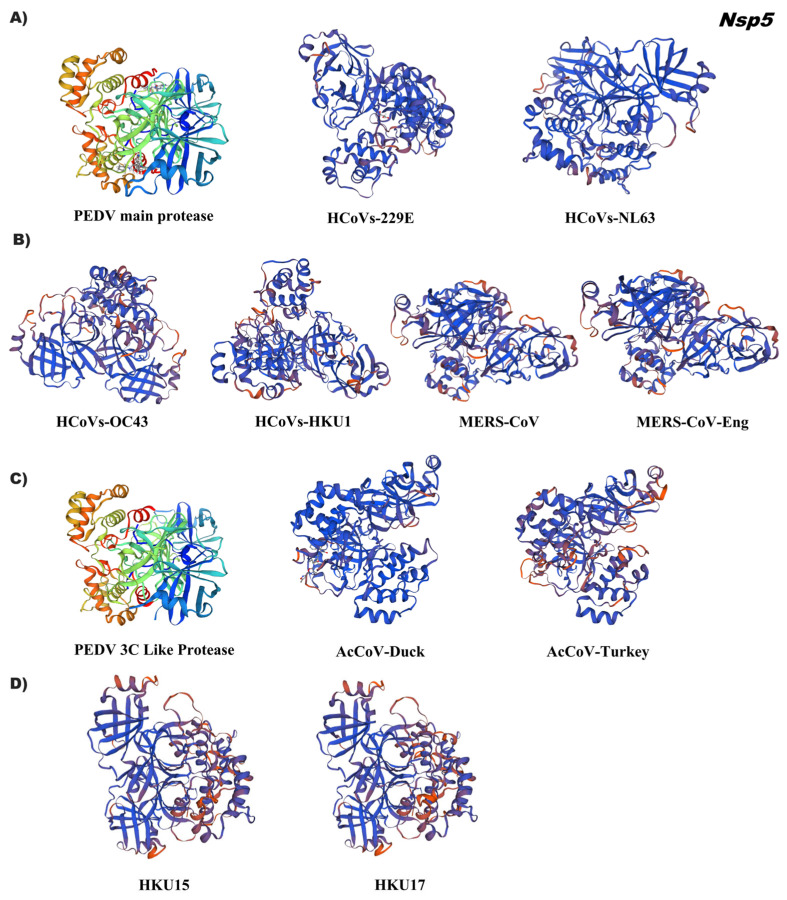
Tertiary structure of the major protease (Mpro) or 3C-like protease (3CLpro) Nsp5 of the coronavirus genus alpha, beta, gamma, and delta. (**A**) Alphacoronavirus: HcoVs-229E (NP_073549.1), HcoVs-NL63 (YP_003766.2); (**B**) Betacoronavirus: HcoVs-OC43 (YP_009924323.1), HcoVs-HKU1(YP_009944273.1), MERS-CoV (YP_009047217.1), MERS-CoV-Eng (YP_009944296.1); (**C**) Gammacoronavirus: AcCoV-Duck (YP_009825033.1), AcCoV-Turkey (YP_001941178.1); (**D**) Deltacoronavirus: Porcine coronavirus HKU15 (QWE80491.1), Sparrow deltacoronavirus HKU17 (AWV67106.1). Reference ID: 6w81.1 (Alphacoronavirus and Betacoronavirus)), 4xFq.1 (Gammacoronavirus and Deltacoronavirus). Tertiary structures were made using SWISS MODEL software.

**Figure 5 pathogens-12-01185-f005:**
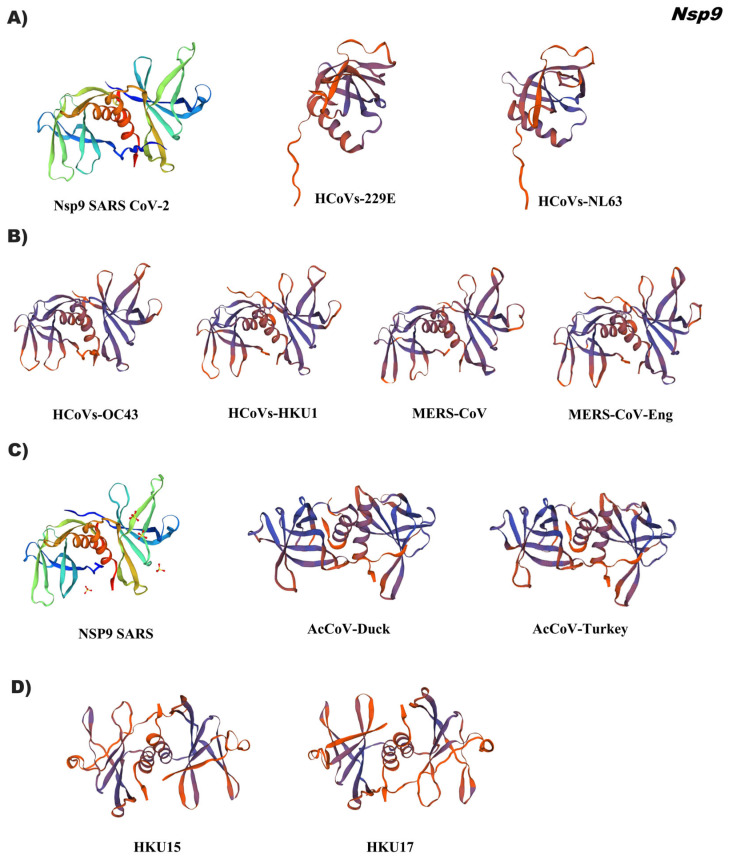
Tertiary structure in silico of Nsp9 of the coronavirus genus alpha, beta, gamma, and delta. (**A**) Alphacoronavirus: HcoVs-229E (NP_073549.1), HcoVs-NL63 (YP_003766.2); (**B**) Betacoronavirus: HcoVs-OC43 (YP_009555238.1), HcoVs-HKU1 (YP_173236.1), MERS-CoV (YP_009047202.1), MERS-CoV-Eng (YP_007188577.3); (**C**) Gammacoronavirus: AcCoV-Duck (YP_009825037.1), AcCoV-Turkey (YP_001941182.1); (**D**) Deltacoronavirus: Porcine coronavirus HKU15 (QWE80491.1), Sparrow deltacoronavirus HKU17 (AWV67106.1). Reference ID: 6w4b.1 (Alphacoronavirus, Betacoronavirus, and Deltacoronavirus)), Iqz8.1 (Gammacoronavirus). Tertiary structures were made using SWISS MODEL software.

**Figure 6 pathogens-12-01185-f006:**
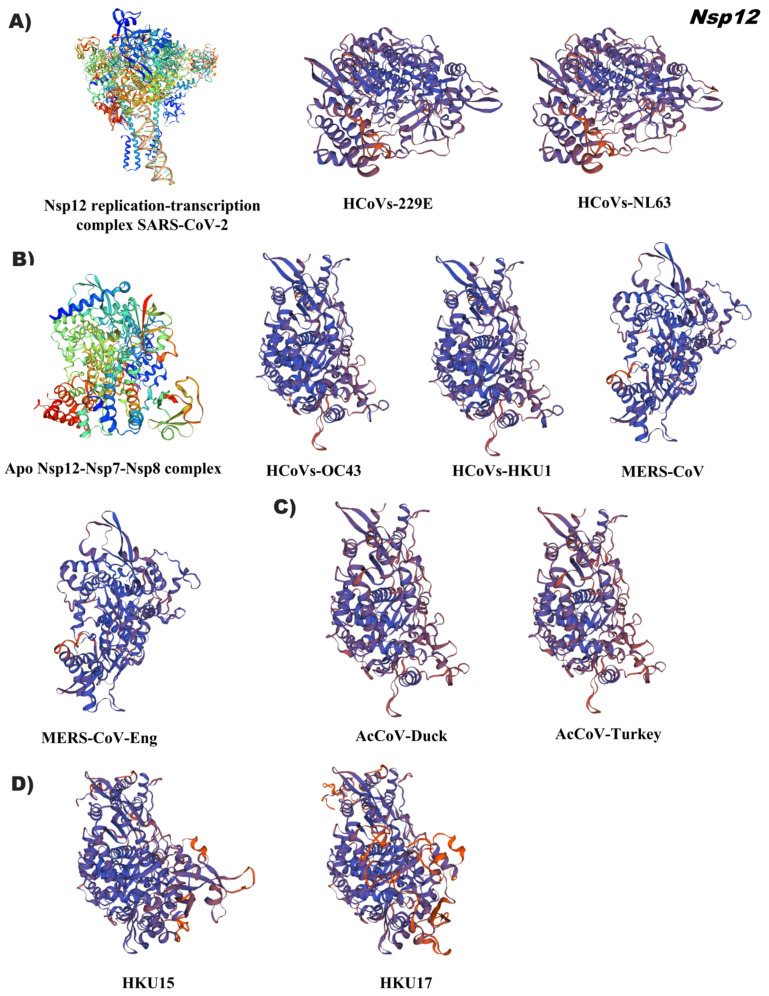
Tertiary structure of Nsp12 of the coronavirus genus alpha, beta, gamma, and delta. (**A**) Alphacoronavirus: HCoVs-229E (NP_073549.1), HCoVs-NL63 (YP_003766.2); (**B**) Betacoronavirus: HCoVs-OC43 (YP_009555260.1), HCoVs-HKU1 (YP_459941.1), MERS-CoV (YP_009047223.1), MERS-CoV-Eng (YP_009944302.1); (**C**) Gammacoronavirus: AcCoV-Duck (YP_009825029.1), AcCoV-Turkey (YP_001941185.1); (**D**) Deltacoronavirus: Porcine coronavirus HKU15 (QWE80491.1), Sparrow deltacoronavirus HKU17 (AWV67106.1). Reference ID: 6xez.1 (Alphacoronavirus, Gammacoronavirus and Deltacoronavirus-HKU15)), 7bu1.1 (Betacoronavirus and Deltacoronavirus-HKU17). Tertiary structures were made using SWISS MODEL software (https://swissmodel.expasy.org/) (accessed on 19 February 2023).

**Figure 7 pathogens-12-01185-f007:**
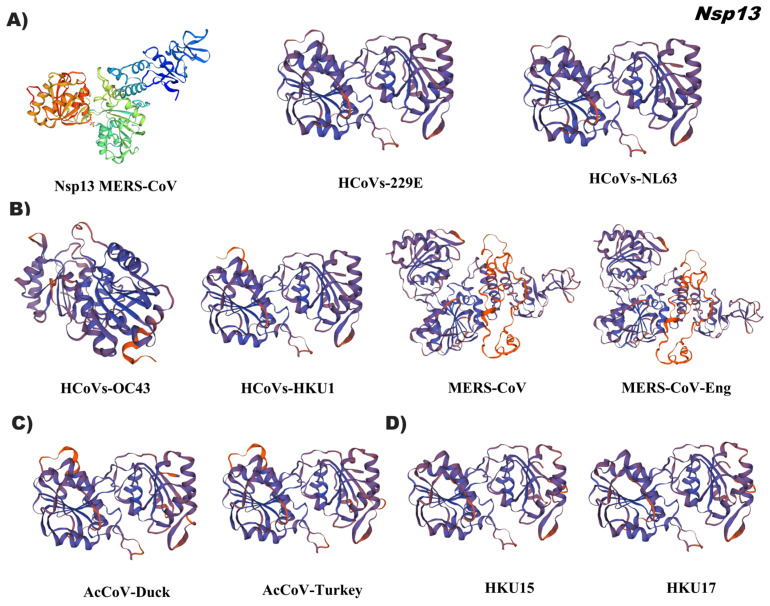
Tertiary structure of Nsp13 of the coronavirus genus alpha, beta, gamma, and delta. (**A**) Alphacoronavirus: HCoVs-229E (NP_073549.1), HCoVs-NL63 (YP_003766.2); (**B**) Betacoronavirus: HCoVs-OC43 (YP_009555238.1), HCoVs-HKU1 (YP_459942.1), MERS-CoV (YP_009047224.1), MERS-CoV-Eng (YP_009944303.1); (**C**) Gammacoronavirus: AcCoV-Duck (YP_009825025.1), AcCoV-Turkey (YP_001941186.1); (**D**) Deltacoronavirus: Porcine coronavirus HKU15 (QWE80491.1), Sparrow deltacoronavirus HKU17 (AWV67106.1). Reference ID: 5wwp.2. Tertiary structures were made using SWISS MODEL software.

**Figure 8 pathogens-12-01185-f008:**
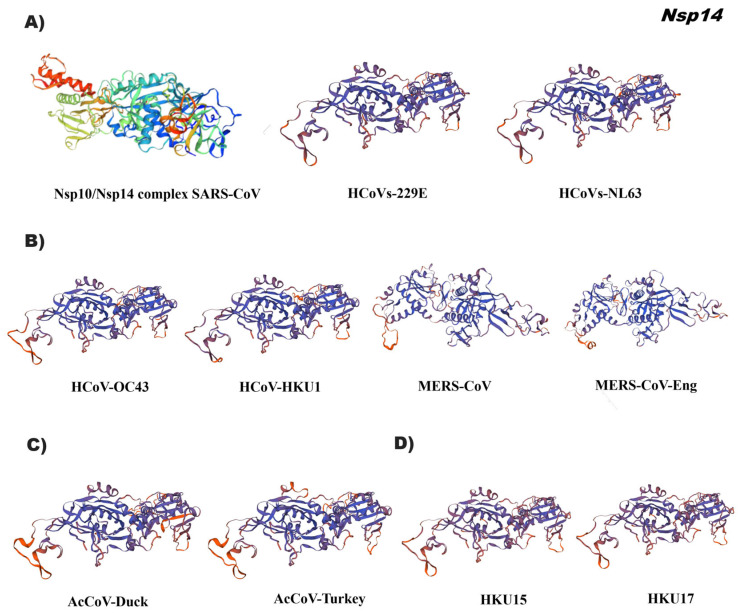
Tertiary structure in silico of Nsp14 of the coronavirus genus alpha, beta, gamma, and delta. (**A**) Alphacoronavirus: HCoVs-229E (NP_073549.1), HCoVs-NL63 (YP_003766.2); (**B**) Betacoronavirus: HCoVs-OC43 (YP_009555238.1), HCoVs-HKU1 (YP_173236.1), MERS-CoV (YP_009047202.1), MERS-CoV-Eng (YP_007188577.3); (**C**) Gammacoronavirus: AcCoV-Duck (YP_009825006.1), AcCoV-Turkey (YP_001941164.2); (**D**) Deltacoronavirus: Porcine coronavirus HKU15 (QWE80491.1), Sparrow deltacoronavirus HKU17 (AWV67106.1). Reference ID: 5wwp.2. Tertiary structures were made using SWISS MODEL software.

**Figure 9 pathogens-12-01185-f009:**
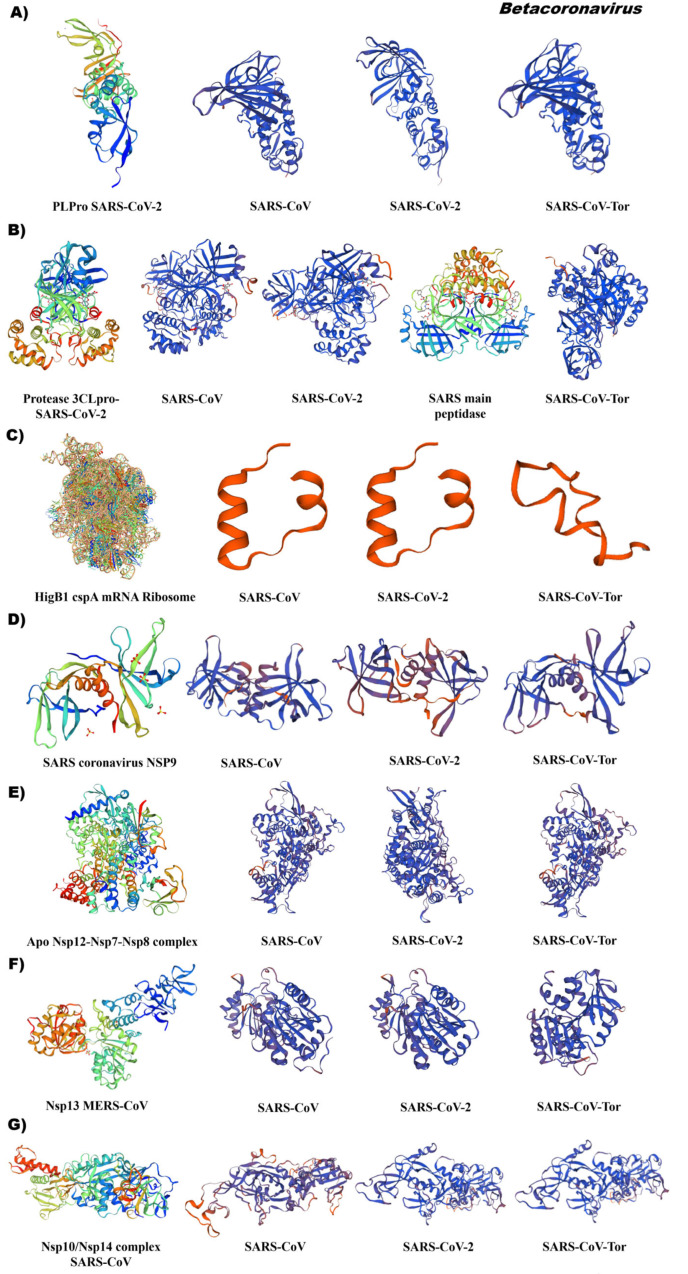
Tertiary structure of non-structural proteins (Nsp) of the genus betacoronavirus. (**A**) Nsp 3; (**B**) Nsp5, (**C**) Nsp6, (**D**) Nsp9, (**E**) Nsp12, (**F**) Nsp13, (**G**) Nsp14. Accessions: SARS-CoV (APO40578.1), SARS-CoV-2 (YP_009724389.1), SARS-CoV-Tor2 (NP_828849.7). Reference ID Nsp3: 7d6h.1 (Papain-like protease PLPro SARS-CoV-2); Nsp5 7dpp.1 (Protease 3CLpro-SARS-CoV-2) and 2a5i.1 (SARS main peptidase); Nsp6: 7nbu.1 (HigB1 cspA mRNA Ribosome); Nsp9: 1qz8.1 (SARS coronavirus NSP9); Nsp12: 7bv1.1 (Apo Nsp12-Nsp7-Nsp8 complex); Nsp13: 5nfy.2 (Nsp10/Nsp14 dynamic complex SARS-CoV). Tertiary structures were made using SWISS MODEL software.

**Figure 10 pathogens-12-01185-f010:**
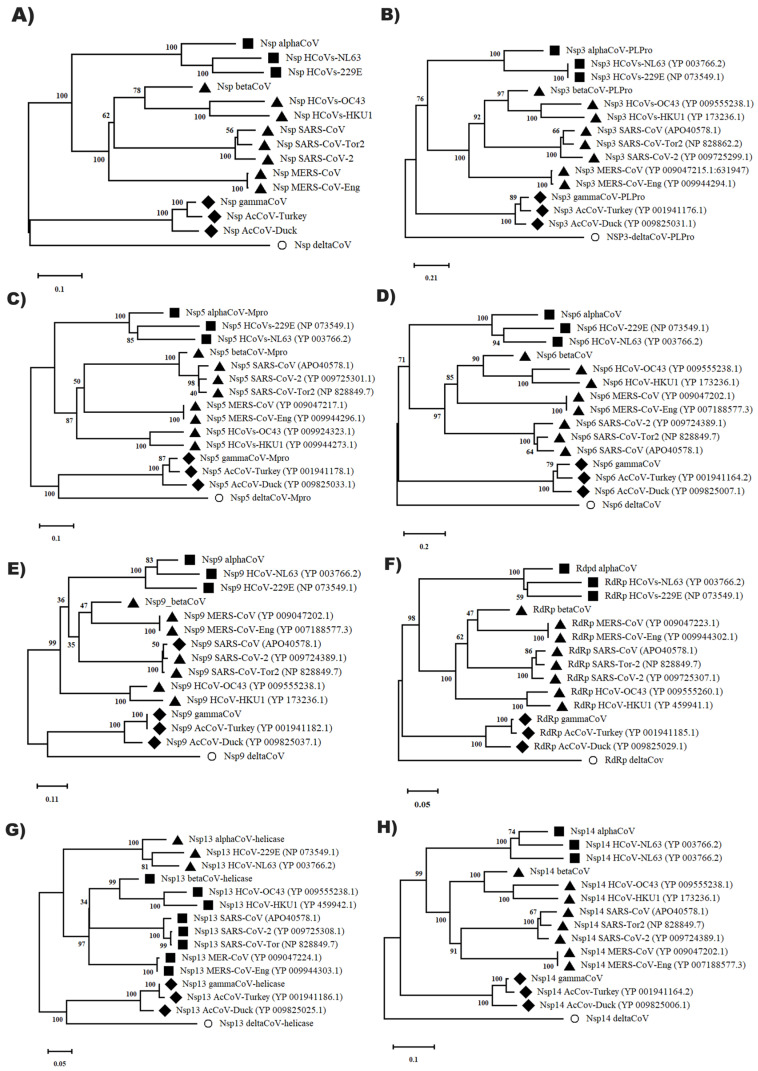
Evolutionary relationships of structural proteins (Nsp) of human pathogenic coronaviruses. (**A**) Cluster Nsp; (**B**) PLPro (Nsp3); (**C**) Mpro (Nsp5); (**D**) Nsp6; (**E**) Nsp9; (**F**) Rdpd (Nsp12); (**G**) Nsp13 helicase; (**H**) Nsp14. The evolutionary relationships were made by the MEGA v.11 program with the method of minimal evolution and with a bootstrap of 1000 repetitions. The partial sequences were greater than 100 amino acid residues of different accessions and DeltaCoV was used as an external group.

**Table 1 pathogens-12-01185-t001:** Families of structural proteins of the coronavirus pangenome.

Family/Function	Protein Coding Gene *	Genome
Hemagglutinin-esterase glycoprotein	*gp02*	HCoVs-229E HCoVs-NL63 HCoVs-HKU1 MERS-CoV MERS-CoV-Eng SARS-CoV SARS-CoV-2 SARS-CoV-Tor2 AcCoV- Turkey Porcine coronavirus HKU15
Spike protein	*gp03*	HCoVs-OC43 HCoVs-HKU1 AcCoV-Duck Porcine coronavirus HKU15 Sparrow coronavirus HKU17
Spike surface glycoprotein	*gp04*	HCoVs-OC43
	*gp06*	HCoVs-NL63
		Porcine coronavirus HKU15
		Sparrow coronavirus HKU17
	*gp07*	HCoVs-229E HCoVs-HKU1
	*gp11*	AcCoV-Duck
		AcCoV- Turkey
Nucleocapsid phosphoprotein	*gp08*	HCoVs-OC43 HCoVs-HKU1
Envelope protein	*gp03*	Porcine coronavirus HKU15 Sparrow coronavirus HKU17
	*gp05*	HCoVs-229E HCoVs-NL63 HCoVs-HKU1
	*gp06*	AcCoV-Duck AcCoV- Turkey
Membrane protein	*gp04*	HCoVs-NL63 Porcine coronavirus HKU15
		Sparrow coronavirus HKU17
	*gp06*	Human coronavirus 229E Human coronavirus HKU1
	*gp07*	HCoVs-OC43 AcCoV-Duck AcCoV- Turkey

* Gene locus encoding coronavirus structural proteins.

**Table 2 pathogens-12-01185-t002:** Functional annotation and identification of non-structural protein (Nsp) domains in the coronavirus genome.

Protein Family Domain (CDD)	Alphacoronavirus (AVA26872.1) *	Betacoronavirus (YP_009724389.1) *	Gammacoronavirus (YP_009825006.1) *	Deltacoronavirus (AWV67106.1) *
*shared domains*				
Non-structural protein 2 (Nsp2) and related proteins	cd21514: similar alphaCoV_Nsp2_HCoV-229E	cd21516: betaCoV_Nsp2_SARS-like	cl40697: cv_gamma-delta_Nsp2_IBV-like Superfamily	cd21512: cv_gamma-delta_Nsp2_IBV-like
Non-structural protein 4 transmembrane domain (Nsp4)	cd21473: cv_Nsp4_TM	cd21473: cv_Nsp4_TM	cd21473: cv_Nsp4_TM	cd21473: cv_Nsp4_TM
Coronavirus non-structural protein 4 C-terminal (Nsp4)	pfam16348: Corona_NSP4_C	pfam16348: Corona_NSP4_C	cl24800: Corona_NSP4_C Superfamily	cl24800: Corona_NSP4_C Superfamily
Non-structural protein 5 (Nsp5), major protease (Mpro)	cd21665: alphaCoV_Nsp5_Mpro	cd21666: betaCoV_Nsp5_Mpro	cd21667: gammaCoV_Nsp5_Mpro	cd21668: deltaCoV_Nsp5_Mpro
Non-structural protein 6 (Nsp6)	cd21558: alphaCoV-Nsp6	cd21560: betaCoV-Nsp6	cd21559: gammaCoV-Nsp6	cd21561: deltaCoV-Nsp6
Non-structural protein 8 (Nsp8)	cd21830: alphaCoV_Nsp8	pfam08717: nsp8	cd21832: gammaCoV_Nsp8	cd21833: deltaCoV_Nsp8
Non-structural protein 10 (Nsp10)	cd21901: alpha_betaCoV_Nsp10	cd21901: alpha_betaCoV_Nsp10	cd21902: gammaCoV_Nsp10	cd21903: deltaCoV_Nsp10
RNA-dependent RNA polymerase (RdRp) of coronaviruses	cd21588: alfaCoV_RdRp	cd21591: SARS-CoV-like_RdRp	cd21587: gammaCoV_RdRp	cd21590: deltaCoV_RdRp
Helicase domain of non-structural protein 13 (Nsp13)	cd21723: alphaCoV_Nsp13-helicase	cd21722: betaCoV_Nsp13-helicase	cd21720: gammaCoV_Nsp13-helicase	cd21721: deltaCoV_Nsp13-helicase
Stem domain of Nsp13 helicase and related proteins	cd21689: stalk_CoV_Nsp13-like	cd21689: stalk_CoV_Nsp13-like	cd21689: stalk_CoV_Nsp13-like	cd21689: stalk_CoV_Nsp13-like
Cys/His-rich zinc-binding domain (CH/ZBD) helicase Nsp13 of the coronavirus SARS and related proteins	cd21401: ZBD_cv_Nsp13-like	cd21401: ZBD_cv_Nsp13-like	cd21401: ZBD_cv_Nsp13-like	cd21401: ZBD_cv_Nsp13-like
Non-structural protein 14 (Nsp14)	cd21660: alphaCoV_Nsp14	cd21659: betaCoV_Nsp14	cd21658: gammaCoV_Nsp14	cd21657: deltaCoV_Nsp14
N-terminal domain of Non-structural protein 15 (Nsp15) and related proteins	cd21171: NTD_alpha_betaCoV_Nsp15-like	cd21171: NTD_alpha_betaCoV_Nsp15-like	cd22650: NTD_gammaCoV_Nsp15-like	cd21172: NTD_deltaCoV_Nsp15-like
Papain-like protease (PLpro)	cd21731: alphaCoV_PLPro	cd21732: betaCoV_PLPro	cd21733: gammaCoV_PLPro	cd21734: deltaCoV_PLPro
*specific domains*				
Non-structural protein 1 (Nsp1)	cd21875: PEDV-like_alphaCoV_Nsp1	cd21796: SARS-CoV-like_Nsp1_N	nd	nd
C-terminal domain Nsp1 of coronavirus and betacoronaviruses SARS-related in lineage B	nd	cd22662: SARS-CoV-like_Nsp1_C	nd	nd
Coronavirus replicase Nsp2, C-terminal	pfam19212: CoV_NSP2_C	nd	nd	nd
Coronavirus 2′-O-methyltransferase	nd	pfam06460: CoV_Methyltr_2	nd	nd
Pneumococcal surface protein PspC, LPXTG-anchored form	nd	nd	cl41463: PspC_subgroup_2 Superfamily	nd
Domain X (or Mac1 domain) of viral non-structural protein 3 and related macrodomains	cd21557: Macro_X_Nsp3-like	nd	cd21557: Macro_X_Nsp3-like	cd21557: Macro_X_Nsp3-like
C-terminal non-structural protein 3 (Nsp3) including E, Y transmembrane domains	cd21712: TM_Y_alphaCoV_Nsp3_C	cd21717: TM_Y_SARS-CoV-like_Nsp3_C	cd21710: TM_Y_gammaCoV_Nsp3_C	cd21711: TM_Y_deltaCoV_Nsp3_C
Nucleic acid-binding domain of the Nsp3 coronavirus and severe acute respiratory syndrome (SARS)-related betacoronaviruses in the B lineage	nd	cd21822: SARS-CoV-like_Nsp3_NAB	nd	nd
Betacoronavirus-specific marker of SARS-related coronavirus Nsp3 and B-lineage betacoronavirus	nd	cd21814: SARS-CoV-like_Nsp3_betaSM	nd	nd
SARS C-terminal (SUD) single domain of Nsp3 of SARS coronavirus and related betacoronaviruses in the B lineage	nd	cd21525: SUD_C_SARS-CoV_Nsp3	nd	nd
Papain-like protease (PLPro) found in non-structural protein 3 (Nsp3)	cl40457: CoV_PLPro Superfamily	nd	nd	nd
First ubiquitin-like domain (Ubl) located at the N-terminus of SARS-CoV coronavirus Nsp3 and related proteins	nd	nd	cl28922: Ubl1_cv_Nsp3_N-like Superfamily	nd
Single-stranded poly(A) binding domain of Nsp3		cl13138: SUD m Superfamily		
Non-structural protein 7 (Nsp7)	cd21826: alphaCoV_Nsp7	cd21827: betaCoV_Nsp7	nd	cd21829: deltaCoV_Nsp7
Non-structural protein 9 (Nsp9)	cd21897: alphaCoV_Nsp9	nd	cd21899: gammaCoV_Nsp9	cd21900: deltaCoV_Nsp9
mRNA cap-1 methyltransferase from Nsp13	cl20156: NSP13 Superfamily	nd	cl20156: NSP13 Superfamily	cl20156: NSP13 Superfamily
Domain 1B of NSP13 helicase and related proteins of the SARS coronavirus	cd21409: 1B_cv_Nsp13-like	nd	nd	cd21409: 1B_cv_Nsp13-like
Nidoviral uridylate-specific endoribonuclease (NendoU) domain of non-structural protein 15 (Nsp15) and related proteins	cd21161: NendoU_cv_Nsp15-like	cd21161: NendoU_cv_Nsp15-like	nd	cd21161: NendoU_cv_Nsp15-like
Middle domain of Non-structural Protein 15 (Nsp15) and related proteins	cd21167: M_alpha_beta_cv_Nsp15-like	nd	cd21168: M_gcv_Nsp15-like	cd21169: M_dcv_Nsp15-like
Macrodomain superfamily	nd	cl00019: Macro_SF Superfamily	nd	nd

* Label domains found in the Pfam, SMART, COG, PRK, TIGRFAM and CDD databases. nd. Not determined.

## Data Availability

Not applicable.

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
