# Peer review of "Non-Structural Proteins (Nsp): A Marker for Detection of Human Coronavirus Families"

_pathogens, 2023, doi:10.3390/pathogens12091185_

Round 1

Reviewer 1 Report

In the manuscript entitled ¨Nonstructural proteins (Nsp) a marker for detection of human 2 coronavirus families¨ the authors declare ¨In this research, we performed functional annotation and identification of changes in (non-structural proteins) Nsp domains in the coronavirus genome¨. More relevant is the last sentence of the abstract, ¨These types of evolutionary studies provide a window of opportunity to use 33 these Nsp as targets of viral therapies¨. Nevertheless, this last part of the abstract is scarcely discussed. Other issues need to be improved.

1. Erase the text in 156 to 170 because is related to journal instructions. The same for lines 199 to 201.

2. Considering the extension of the figures, it is necessary to improve the quality of the images

3. In Fig 2 to 10, include the software used in the footnote.

4. In line 728, Clarify if the authors refer to the genus in the phylogenetic aspect or a gender-biomarker (female or male).

5. In the figure captions, include the bold letters that correspond.

The author needs to improve the discussion related to the relevance of the site that could be used for new drug development, this is the relevancy of all in silico analysis performed by the authors.

The manuscript needs minor English edition 

Author Response

Dear Reviewer:

Receive in the most attentive way letter of response of the requested revision. Please see the attachment.

Reviewer 2 Report

The work is very well done and contributes to the field of knowledge. The evaluation of evolutionary studies provides an opportunity to use non-structural proteins (Nsp) as targets for viral therapies, as is currently being scientifically advanced. In addition, evolutionary markers . Also demonstrates the identification of evolutionary markers that allow us to follow their evolutionary trajectory and to know the origin of the genetic divergence lineage that impacts their viral classification. as well as to search for biomedical technology for their application in the health field. The authors  discuss the results, which can be interpreted from the perspective of previous studies and working hypotheses. The results and their implications are discussed in the broadest possible context. Future research directions such as applicability . It also shows that changes in the 3D structure of the Nsp suggest adaptation of coronaviruses during their viral infection mechanism. Another interesting point is that Table S1 shows the main types of coronavirus that infect humans

Author Response

(The authors gave the same response as above.)

Reviewer 3 Report

This manuscript titled “Nonstructural proteins (Nsp) a marker for detection of human coronavirus families” performed a functional annotation and identification of changes in (non-structural proteins) Nsp domains in the coro-navirus genome. The manuscript is well-organized and has certain significance. It was addressed a specific gap in the field, the references are appropriate,

However, there are still some problems in this manuscript that need to be revised;

1.          The language needs considerable attention.

2.          Most of the picture and table quality is generally poor.

Author Response

(The authors gave the same response as above.)

Round 2

Reviewer 1 Report

The authors consider all reviewer suggestions. The manuscript is ready to be published.

Reviewer 3 Report

I have no other comments